# Long-lasting field-free alignment of large molecules inside helium nanodroplets

Adam S. Chatterley[1], Constant Schouder[2], Lars Christiansen[1], Benjamin Shepperson[1], Mette Heidemann Rasmussen[1] & Henrik Stapelfeldt[1]

Molecules with their axes sharply confined in space, available through laser-induced alignment methods, are essential for many current experiments, including ultrafast molecular imaging. For these applications the aligning laser field should ideally be turned-off, to avoid undesired perturbations, and the strong alignment should last long enough that reactions and dynamics can be mapped out. Presently, this is only possible for small, linear molecules and for times less than 1 picosecond. Here, we demonstrate strong, field-free alignment of large molecules inside helium nanodroplets, lasting >10 picoseconds. One-dimensional or three-dimensional alignment is created by a slowly switched-on laser pulse, made field-free through rapid pulse truncation, and retained thanks to the impeding effect of the helium environment on molecular rotation. The opportunities field-free aligned molecules open are illustrated by measuring the alignment-dependent strong-field ionization yield of dibromothiophene oligomers. Our technique will enable molecular frame experiments, including ultrafast excited state dynamics, on a variety of large molecules and complexes.

---

[1] Department of Chemistry, Aarhus University, 8000 Aarhus C, Denmark. [2] Department of Physics and Astronomy, Aarhus University, 8000 Aarhus C, Denmark. Correspondence and requests for materials should be addressed to H.S. (email: henriks@chem.au.dk)

Molecules with their axes sharply aligned in space provide two unique advantages. Firstly, they make it possible to explore or exploit the orientational dependence of molecular interactions, and secondly, they maximize the information content of experimental observables because the usual blurring effect from randomly oriented molecules is minimized. Therefore, aligned molecules are useful for many current and emerging applications in molecular science including time-resolved structural imaging with femtosecond or attosecond techniques[1–4].

Molecular alignment can be obtained through laser-induced methods, relying on the interaction between a moderately intense laser field and the anisotropic polarizability of gas-phase molecules[5,6]. If the laser field is switched on slowly as compared to the rotational period(s) of the molecules, it is possible to obtain one-dimensional (1D) alignment, where the most polarizable molecular axis is fixed in space, by a linearly polarized field or three-dimensional (3D) alignment[7], and where all three principal polarizability axes are fixed in space with an elliptically polarized field. In this (quasi) adiabatic regime the degree of alignment can be very high, but it occurs while the laser pulse is on. This is a serious obstacle for many applications of aligned molecules because the alignment field can perturb the molecules severely. An important example is molecules in electronically excited states, where the alignment field may cause further excitation, dissociation or ionization. Thus, the excited state reactions originally expected may not be observed because they are outcompeted by processes induced by the alignment field[8,9].

Consequently, there have been intense efforts to develop methods that create aligned molecules after the alignment laser field is turned off. The most widespread technique relies on a short, with respect to the molecular rotational period(s), laser pulse (typically a few hundred femtoseconds long) to form a rotational wave packet. The wave packet leads to periodic alignment in narrow time windows, termed revivals, long after the pulse is turned off[5]. For linear or symmetric top molecules, the degree of 1D alignment can be high during the revivals and, thereby, offers an opportunity to conduct molecular frame experiments under field-free conditions, which has been exploited in a range of studies[1,3,4,10,11]. The fast rotational dispersion characteristic of freely rotating gas-phase molecules limits, however, the duration of the revivals to typically less than a picosecond. This precludes molecular frame observations of most inter- and intramolecular processes for their entire duration. Field-free 3D alignment suffers from the same limitation and is further complicated by the irregular rotational level structure of asymmetric top molecules, which tends to reduce the obtained degree of alignment[12–15].

Here we show that it is possible to create strongly aligned molecules inside helium nanodroplets under long-lasting, field-free conditions. Alignment is induced in the quasi-adiabatic limit by a slowly turned-on pulse. At its peak, where the degree of alignment is highest[16,17], the pulse is rapidly truncated. We show that unlike the gas-phase case, where such truncation leads to fast loss of alignment[18,19], the molecules inside He droplets remain aligned for 10 ps or longer owing to the impeding effect of the He environment on molecular rotation observed earlier[17,20,21]. This period should be enough time for many potential dynamics processes. The technique works for 1D and 3D alignment, and is applicable to large and complex molecules, which we demonstrate by 3D alignment of dibromoterthiophene and tetrabromoindigo molecules. To test the field-free nature of the alignment, we measure the time-dependent yield of intact molecular parent ions, formed by a femtosecond probe pulse, because they only survive fragmentation by the alignment field if it is sufficiently weak. As the alignment pulse turns off, we observe that the parent ion

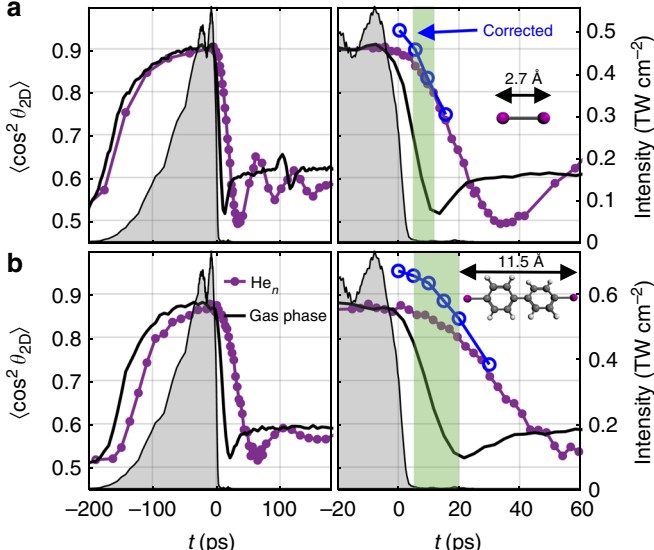

**Fig. 1** 1D alignment of $I_2$ and diiodobiphenyl. Alignment dynamics of $I_2$ (**a**) and 4,4′-diiodobiphenyl (**b**) molecules, represented by $\langle \cos^2 \theta_{2D} \rangle$ and recorded as a function of time. The measurements are conducted both for isolated molecules (black curves) and for molecules in He droplets (purple curves) under identical laser conditions. The intensity profile of the truncated alignment pulse is shown by the grey shaded area and refers to the right vertical axis. The panels on the right show a zoom of the post-truncation region. The green shaded area in each panel marks the interval where the alignment field intensity is <1% of its peak and $\langle \cos^2 \theta_{2D} \rangle \geq 0.80$. The structure of each molecule is inset, along with a scalebar representing the I–I distance. In droplets, the values for $\langle \cos^2 \theta_{2D} \rangle$ after correction for non-axial recoil at selected times are also shown as blue open circles

signal increases steeply from zero to a maximum occurring when the degree of alignment is still high, demonstrating strong alignment under conditions where the alignment field has been reduced enough that its perturbing effect is negligible. Finally, we demonstrate the potential of our technique for molecular frame experiments by performing a prototypical alignment-dependent strong-field ionization experiment on dibromoterthiophene molecules, which would not be possible in the presence of an alignment laser field.

## Results

**One-dimensional field-free alignment.** The first step of our technique is to induce strong molecular alignment with a laser pulse that is turned on slowly (10–90% intensity in 100 ps, $\lambda_{centre} = 800$ nm, peak intensity $I_{align}$ ~$6 \times 10^{11}$ W cm$^{-2}$). The temporal evolution of the alignment is measured through timed Coulomb explosion, triggered by an intense 40 fs probe pulse sent at time $t$, and recording of the emission direction of fragment ions (see Methods). The purple curve in Fig. 1a depicts the result for 1D alignment of iodine molecules ($I_2$, polarizabilities of all molecules are given in Supplementary Table 1) inside He droplets using a linearly polarized alignment pulse. The degree of alignment, characterized by the expectation value $\langle \cos^2 \theta_{2D} \rangle$, rises from 0.5, the value characterizing random alignment, before the pulse to 0.90 at the peak of the pulse, with $\theta_{2D}$ denoting the angle between the projection of the recoil direction of an $I^+$ fragment ion and the polarization of the alignment pulse[16]. For $I_2$ molecules perfectly aligned along the alignment pulse polarization $\langle \cos^2 \theta_{2D} \rangle$ would equal 1. The gradual increase of $\langle \cos^2 \theta_{2D} \rangle$ with the alignment pulse intensity shows that the alignment process essentially evolves adiabatically in accordance with previous experiments[16].

The second step is rapid truncation of the laser pulse at its peak to create field-free alignment, implemented by spectral truncation of a chirped pulse with a longpass optical filter[19]. The alignment laser intensity drops by more than a factor of 100 over ~10 ps; however, the degree of alignment decreases at a much slower rate. The right panel of Fig. 1a shows that $\langle \cos^2 \theta_{2D} \rangle$ retains a value between 0.86 and 0.80 from $t = 5$ to 11 ps. In this 6-ps-long time window, marked by the shaded green area, the alignment pulse intensity is <1% of the value at the peak and the molecules are still well aligned. By comparison, isolated molecules in a cold molecular beam attain a similar value of $\langle \cos^2 \theta_{2D} \rangle$ at the peak of the laser pulse, but after truncation $\langle \cos^2 \theta_{2D} \rangle$ drops much faster and in the 5–11 ps interval $\langle \cos^2 \theta_{2D} \rangle$ is reduced to 0.69–0.52 (black curve in Fig. 1a). The rapid decrease of $\langle \cos^2 \theta_{2D} \rangle$ in isolated $I_2$ is due to rotational dispersion characterizing freely rotating molecules[18,19]. By contrast, in the droplets the impeding effect of the He environment on the molecular rotation[17,20,21] comes to our advantage in terms of granting a period of field-free alignment after the pulse is switched off. Note that this effect cannot be explained solely by either the low temperature of the droplets or an increase of effective moment of inertia for solvated molecules (see Supplementary Note 2). We suspect it is related to a significant deviation of the rotational level structure from a rigid rotor structure. A theoretical description of the dynamics of laser-induced molecular alignment in helium droplets is being explored and developed[16,22,23]; however, this is not required to take advantage of the aligned molecules.

An additional advantage of molecules in He droplets is that they have a rotational temperature of only 0.4 K[24]. In general, this leads to stronger (quasi-)adiabatic alignment compared to gas-phase molecules[16] because the latter typically cannot be cooled to such low temperatures by supersonic jet-cooling techniques[25–27]. In fact, the true degree of alignment is better than the $\langle \cos^2 \theta_{2D} \rangle$ values stated so far because the initial recoil direction of the $I^+$ ions, defined by the alignment of their parent molecule, is blurred due to collisions with He atoms on the way out of the droplet towards the detector. This effect can be characterized by measuring deviations from perfect back-to-back axial recoil using angular covariance mapping[28]. Correcting for this effect, we find that $\langle \cos^2 \theta_{2D} \rangle$ is actually between 0.90 and 0.84, shown as blue circles in Fig. 1, in the 5–11 ps field-free interval.

Next, we turn to a much larger molecule, 4,4′-diiodobiphenyl (DIBP, inset Fig. 1b). The linearly polarized laser pulse is expected to align the most polarizable molecular axis, which is the I–I axis, along its polarization axis in a situation similar to that of the $I_2$ molecules. Again, $\langle \cos^2 \theta_{2D} \rangle$ is obtained from recording the emission direction of the Coulomb exploding $I^+$ ions. The results, displayed in Fig. 1b, show that for DIBP in He droplets $\langle \cos^2 \theta_{2D} \rangle$ reaches 0.87 (0.95) shortly before truncation and 0.86–0.80 (0.94–0.84) between $t = 5$ and 20 ps, where the laser intensity is reduced to <1% of its maximum value. (The numbers within parentheses are the $\langle \cos^2 \theta_{2D} \rangle$ values obtained after correction for non-axial recoil effects and scattering on He atoms[16,28].) For the isolated DIBP molecules such an interval of strong, field-free alignment is not present. Compared to $I_2$, DIBP has a much higher moment of inertia, and the degree of alignment decays slower after pulse truncation. This points to the effect that, in general, the larger the molecule is, the longer it can be field-free aligned in the He droplets (compare the shaded green areas in Fig. 1a, b).

**3D field-free alignment**. To demonstrate the generality of our technique, notably towards complex systems, we performed experiments on two more molecular species embedded in He

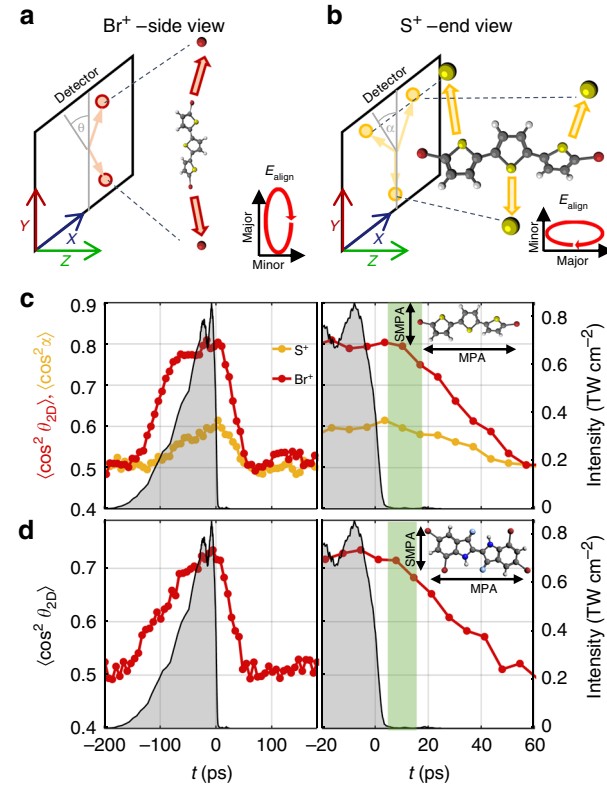

**Fig. 2** 3D alignment of dibromoterthiophene and tetrabromoindigo. **a**, **b** Illustration of how 3D alignment of DBT is characterized. In **a** the molecule is in a side view with the most polarizable axis (MPA) confined to the major polarization axis of the alignment pulse, directed along the $Y$-axis. In **b** the molecule is in an end view, with the MPA and major polarization axis directed on the $Z$-axis. In the side view, alignment is characterized by $\langle \cos^2 \theta_{2D} \rangle$, where $\theta_{2D}$ is the angle between the emission direction of a $Br^+$ ion and the $Y$-axis, while in the end view it is characterized by $\langle \cos^2 \alpha \rangle$, where $\alpha$ is the angle between the emission direction of a $S^+$ ion and the $Y$-axis. **c** The time dependence of $\langle \cos^2 \theta_{2D} \rangle$ and $\langle \cos^2 \alpha \rangle$ for DBT molecules induced by an elliptically polarized alignment pulse with an intensity ratio of 3:1. **d** The time dependence of $\langle \cos^2 \theta_{2D} \rangle$ for TBI molecules induced by an elliptically polarized alignment pulse with an intensity ratio of 3:1. These data were recorded in the side view, and as for DBT, $\theta_{2D}$ refers to the angle between the MPA and the major polarization axis. For both **c**, **d** the right vertical axis gives the intensity of the alignment pulse. The panels on the right show a zoom of the post-truncation region and the green shaded areas mark the field-free aligned intervals where $\langle \cos^2 \theta_{2D} \rangle$ has dropped by <10%. The MPA and SMPA are overlaid on the molecular structures

droplets. The first, 5,5″-dibromo-2,2′:5′,2″-terthiophene (DBT, inset Fig. 2c), is an oligomer of three thiophene units and can be thought of as a prototype of polythiophenes used in molecular electronics[29]. The alignment pulse is elliptically polarized with the purpose of inducing 3D alignment[7]. Therefore, we expect that the most polarizable axis (MPA) aligns along the major polarization axis simultaneously with the second MPA (SMPA) aligning along the minor polarization axis, see Fig. 2a, b and Supplementary Note 1. The alignment dynamics recorded for DBT, shown in Fig. 2c, confirm these expectations. Around the peak of the pulse $\langle \cos^2 \theta_{2D} \rangle$ ($\langle \cos^2 \alpha \rangle$) for the $Br^+$ ($S^+$) ions reaches a maximum of ~0.80 (~0.60), showing that the MPA and the SMPA are aligned simultaneously and, therefore, that the molecule is 3D aligned (see Supplementary Figs. 2 and 3 for details, and Fig. 2b for the definition of $\alpha$). This is consistent with recent demonstrations of 3D alignment for smaller molecules in He droplets[17]. The

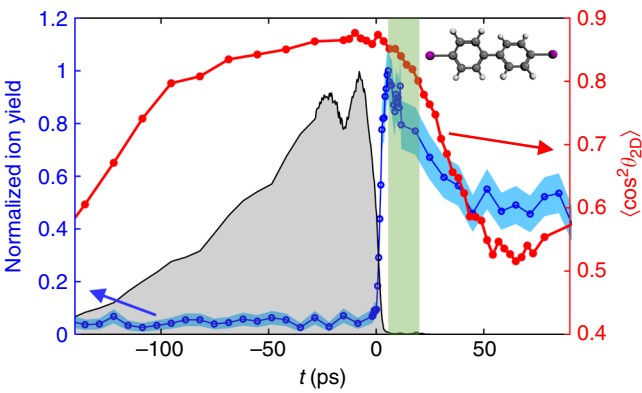

**Fig. 3** Time-dependent yield of intact diiodobiphenyl$^+$ ions from strong-field ionization. The DIBP molecules are 1D aligned. The 800 nm ionization pulse is 40 fs long (full-width at half-maximum (FWHM)), has an intensity of $2.4 \times 10^{14}$ W cm$^{-2}$ and is linearly polarized perpendicular to the polarization axis of the alignment pulse. The blue shaded area represents the 95% confidence intervals for the parent signal, assuming Poisson statistics. The time-dependent degree of alignment is shown by the red curve

$\langle \cos^2 \theta_{2D} \rangle$ and $\langle \cos^2 \alpha \rangle$ values observed here are lower than in our previous demonstration of 3D molecular alignment in He droplets[17], because the fragment ions detected, notably the S$^+$ ions, do not recoil directly along the aligned axes as illustrated in Fig. 2a, b. This is a natural consequence of the complex structure and low symmetry of DBT. So unlike in the cases of I$_2$ and DIBP, $\langle \cos^2 \theta_{2D} \rangle$ can only provide a qualitative rather than a quantitative measure for the degree of alignment.

The crucial finding is, however, the lingering of the alignment after pulse truncation. From $t = 5$ to 24 ps, where the laser intensity is <1% of the peak value, $\langle \cos^2 \theta_{2D} \rangle$ for the Br$^+$ ions drops only from 0.80 to 0.72, and remains >0.58 for the S$^+$ ions. This demonstrates a 19-ps-long interval, marked by the green area, where the molecules are 3D aligned under field-free conditions. The second complex molecule studied was 5, 7, 5′ 7′-tetrabromoindigo (TBI, inset Fig. 3d), a brominated derivative of the indigo dye responsible for the colour in blue jeans. Figure 2b shows that it is also possible to 3D align this molecule for about 17 ps under field-free conditions. Details are given in Supplementary Note 1. Taken together, these results show the generality of our method: We can align a wide range of molecules. As the technique does not depend on rotational revivals, there is no requirement for molecular symmetry other than that the polarizability must be anisotropic.

**Demonstration that alignment is field free**. To assess the field-free nature of the alignment created, we measured the yield of intact parent cations, here created by strong-field ionization with the probe pulse. If the parent ion is created in the presence of an alignment pulse, which is the case in the adiabatic alignment regime, the parent ion will subsequently be exposed to the strong alignment field during its turn-off, which typically lasts >100 ps. While the alignment pulse is non-resonant for the neutral parent molecules, the cations of most large molecules will absorb at the alignment laser wavelength (800 nm). As a consequence, the parent ion is likely to absorb one or several photons from the alignment pulse, leading to dissociation or even further ionization. This destruction of the parent ion has been observed in several previous experiments[8,9] and precluded the use of the parent ions as experimental observables.

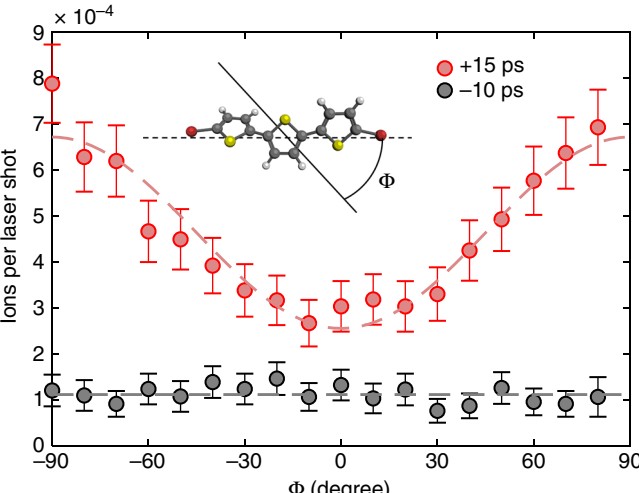

**Fig. 4** Alignment-dependent strong-field ionization of dibromoterthiophene. DBT molecules are 1D aligned by a linearly polarized alignment pulse and ionized by a linearly polarized probe pulse (same parameters as mentioned in the caption to Fig. 3). The yield of intact DBT$^+$ parent ions is measured as a function of the angle, $\Phi$, between the polarization directions of the two laser pulses—illustrated on the sketch of the molecular structure. The measurements are performed at two times, around the peak of the alignment pulse ($t = -10$ ps) and in the field-free window ($t = 15$ ps). The dashed lines serve to guide the eye. Error bars represent the 95% confidence interval, assuming Poisson statistics

Figure 3 shows the yield of parent ions created by ionization of 1D aligned DIBP molecules with a strong-field probe pulse at time $t$. When the probe pulse arrives during the alignment pulse, the parent ion signal is almost zero, but when it arrives after the pulse, the parent ion signal increases sharply and reaches a maximum at $t \sim 7$ ps. The sudden increase of the parent signal by almost a factor of 20 shows that the truncation reduces the alignment pulse intensity sufficiently to prevent destruction of the parent ions. Importantly, the maximum of the parent ion yield occurs at a time where the molecules are still strongly aligned as can be seen from the $\langle \cos^2 \theta_{2D} \rangle$ curve, reproduced on the figure. Figure 3 demonstrates that our method enables ionization experiments on sharply aligned molecules, without the destruction of the fragile parent ions by the alignment field. Note that after the maximum the parent ion yield decreases with the degree of alignment because the strong-field processes involved depend on the molecular alignment[30,31].

Finally, we performed a simple proof-of-principle experiment to demonstrate the possibility to perform measurements within the prolonged field-free alignment period for large molecules. The chosen example was a molecular frame strong-field ionization experiment, an analogue to weak-field linear dichroism experiments[32]. We measured the alignment dependence of the ionization yield of DBT molecules, a phenomenon that has been studied for smaller molecules in the gas phase[30,31,33]. In general, the strong-field ionization probability is a function of the relative angle between the polarization of the ionizing field and the molecular axes. The DBT molecules were 3D aligned and strong-field ionized by a linearly polarized probe pulse sent at $t = 15$ ps. The yield of the DBT$^+$ ions was recorded as a function of the angle between the probe pulse polarization and the major polarization axis of the alignment pulse, which held the MPA fixed in space and approximately coincided with the Br–Br axis (see Fig. 2). The results, represented by the red circles in Fig. 4, show that the detection of intact DBT$^+$ ions more than doubles

when the probe pulse is polarized perpendicular, instead of parallel, to the MPA axis. Our measurement would have been impossible in the presence of the alignment field as illustrated by the black-circle data points recorded at the peak of the alignment pulse ($t = -10$ ps). Here the alignment dependence of the ionization yield is not visible at all. Note that in previous demonstrations of molecular frame strong-field ionization yields, the systems employed were either symmetric molecules amenable to nonadiabatic field-free alignment[30,31,33] or immediately destroyed by the alignment field[31], preventing detection of intact cations.

## Discussion

We have shown that a slowly turned-on, rapidly turned-off laser pulse provides an approach to align a broad class of large molecules inside He nanodroplets under conditions that are effectively field free for a period of tens of picoseconds. We demonstrated that the residual alignment field in this period is so weak that it leaves fragile molecular ions unaffected. The same is expected for molecules in electronically excited states, which opens unexplored possibilities for femtosecond time-resolved imaging of molecules undergoing fundamental photo-induced intramolecular processes in electronically excited states using Coulomb explosion, linear dichroism, or diffraction by ultrashort x-ray or electron pulses[34–37].

An important question is for how long are molecules sufficiently aligned to perform these experiments? This of course depends both on the molecule in question and the degree of alignment required for a particular experiment. As an example, the required degree of alignment for simple molecular frame photoelectron angular distributions was estimated as $\langle \cos^2 \theta \rangle \geq 0.87$ (corresponding to $\langle \cos^2 \theta_{2D} \rangle \geq 0.93$)[38]. Our current set-up with DIPB would give an experimental window of around 4 ps, which is sufficient time to observe photochemical or photophysical unimolecular processes. Even higher degrees of alignment are required for X-ray diffraction experiments, estimated as $\langle \cos^2 \theta \rangle \geq 0.90$ (corresponding to $\langle \cos^2 \theta_{2D} \rangle \geq 0.95$)[39]. In our current experiments, we only reach this degree of alignment during the truncation of the pulse. However, there are several straightforward ways our technique could be extended to give a usable window with these very high degrees of alignment. Firstly, the field-free window could be significantly enhanced by truncating the alignment pulse quicker. Using more advanced pulse shaping schemes, truncation times of 1 ps or less can be obtained[40,41], which would extend the field-free window by around 10 ps. Secondly, the initial degree of alignment, before truncation, can be increased by increasing the intensity of the alignment pulse. For DIBP it should be possible to increase the alignment pulse intensity by a factor of three before ionization sets in. This would result in $\langle \cos^2 \theta_{2D} \rangle \geq 0.97$ and $\langle \cos^2 \theta \rangle \geq 0.95$. Finally, selecting a molecule with a larger moment of inertia will also impede the loss of alignment, as demonstrated in Fig. 1.

We have shown a simple proof-of-principle experiment demonstrating the power of field-free molecular alignment. Although this was a simple demonstration of the ability to perform molecular frame experiments with intact parent ion observables, it opens unexplored possibilities for studying how ionization of large molecules depends on their alignment with respect to the polarization state of the strong laser pulse that induces ionization. This large-molecule regime has not yet been addressed theoretically. Experimentally, the polarization direction of the probe pulse can only by varied in the plane spanned by the most polarizable and the second most polarizable molecular axes defined by the polarization plane of the elliptically polarized alignment pulse. Extending measurements beyond this plane may

be possible by using multiple orthogonally polarized alignment pulses as shown for gas-phase molecules[42–44]. The presence of the helium solvent will influence strong-field physics at sufficiently high intensities of the probe laser pulse. Strong fields have previously been explored for He droplets doped with small clusters of rare gas atoms, in the context of complete ionization of the entire droplets[45,46], and it is intriguing to ask how the He affects strong-field physics at a molecular level.

The technique requires the use of He droplets as a matrix to embed the molecules. Spectroscopic measurements show that helium solvation minimally perturbs electronic and vibrational structure[24], so we anticipate that the reaction dynamics of many molecules will be essentially the same in the droplets as outside. In cases where the reaction dynamics differ, this itself presents an opportunity to study solvation effects, with a weakly perturbing solvent. Additionally, He droplets provide unique opportunities for building molecular complexes[47,48]. There is great potential for applying molecular frame measurements to image such systems and their dynamics. In particular, we have recently developed a scheme for structure determination of complexes of small, sturdy systems by recording parent and fragment ions following Coulomb explosion of sharply aligned complexes[49,50]. We are now able to extend these measurements to dimers of larger fragile molecules, for example, polycyclic aromatic hydrocarbons such as tetracene and pentacene, or even fullerene complexes. As the structure is determined by a femtosecond process, this method should allow real-time observation of structural changes due to intermolecular reactions. For example, the change in structure upon exciplex formation should be visible[51], and we can even envisage using this technique to observe bimolecular reactions[52].

## Methods

The experiment used a helium droplet apparatus[16] and truncated alignment pulses described in detail previously[19]. Helium nanodroplets were produced by continuously expanding 25 bar helium into vacuum through a 5 µm nozzle, cooled to 14 K ($I_2$), 13 K (DIBP and DBT) or 12 K (TBI). Molecules were introduced into the droplets by sending them through a pickup cell filled with molecular vapour, introduced via either a leak valve ($I_2$) or an in-vacuum oven (DIBP, DBT and TBI). In all cases, the vapour pressure in the pickup cell was adjusted to optimize single-molecule doping. The doped helium droplets entered the target region, where they were perpendicularly intersected by two laser pulses. The first, a spectrally truncated, chirped alignment pulse ($\lambda_{centre} = 800$ nm, $\omega_0 = 38$ µm, peak intensity $I_{align} \sim 6 \times 10^{11}$–$9 \times 10^{11}$ W cm$^{-2}$) aligned the molecules. After a time delay controlled by a motorized delay stage, the probe pulse ($\lambda_{centre} = 800$ nm, duration 40 fs full-width at half maximum, $\omega_0 = 25$ µm, $I_{probe} \sim 2.4 \times 10^{14}$ W cm$^{-2}$) multiply ionized molecules, resulting in Coulomb explosion. The probe spot size was considerably smaller than the alignment to minimize focal volume effects. Unless otherwise stated, the alignment laser pulse was polarized with the major axis parallel to the detector, and the probe beam was linearly polarized perpendicular to it.

Degrees of alignment were characterized by Coulomb explosion imaging, and detecting the velocity vectors of charged fragments with a velocity map imaging spectrometer, gated in time such that it was sensitive only to a single ion of interest. Only the ions with higher kinetic energies were considered for $\langle \cos^2 \theta_{2D} \rangle$ determinations, as for these ions the axial recoil approximation is best fulfilled[53], and so they are most representative for measuring $\langle \cos^2 \theta_{2D} \rangle$. Note that this selection procedure does not filter out poorly or unaligned molecules. Parent ion yields were also detected using the velocity map imaging spectrometer, except that only ions with very low kinetic energy were considered. In both cases, there is an unavoidable background from hot effusive molecules of a few per cent, however this does not affect the conclusions. The contribution of these effusive molecules is removed in the non-axial recoil correction shown in Fig. 1[16].

Gas-phase $I_2$ and DIBP molecules were investigated using the same experimental set-up, except that the molecules were sourced from an Even–Lavie pulsed valve, backed by 80 bar of He seeded with $I_2$ or DIBP.

## Data availability

The data measured, simulated and analysed in this study are available from the corresponding author on reasonable request.

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

## Acknowledgements

We acknowledge support from the European Research Council-AdG (Project No. 320459, DropletControl). Also, this research was undertaken as part of the ASPIRE Innovative Training Network, which has received funding from the European Union's Horizon 2020 research and innovation programme under the Marie Sklodowska-Curie grant agreement no. 674960.

## Author contributions

A.S.C., C.S., L.C., B.S. and M.H.R. designed and performed experiments. A.S.C., C.S., L.C. and H.S. analysed the experimental results. All authors took part in regular discussions and were involved in the completion of the manuscript.

## Additional information

**Competing interests:** The authors declare no competing interests.

