## [Peer Review File · Nature Communications]

Reviewers' comments:

Reviewer #1 (Remarks to the Author):

In this manuscript the authors describe the three dimensional (3D) alignment of asymmetric top molecules in a helium (He) droplet under essentially laser-field free conditions. A combination of sophisticated experimental techniques are used to generate and measure the alignment. An elliptically polarized switched laser pulse adiabatically aligns axes of largest and intermediate polarizability of the molecules via rotational Raman scattering. As the pulse abruptly switches off, the molecules are free to rotate again and the alignment decays, precluding field free alignment. This problem was highlighted by Underwood et al. (PRL 97 173001), resulting in this method being previously abandoned as a route to field free three dimensional alignment. The authors have very clearly demonstrated here that in the case of a molecule trapped in a He droplet, not only is this method successful, but can also provide 3D alignment under field free conditions for tens of picoseconds as a result of coupling between the rotations of the molecule to its solvation shell within the droplet. From the point view of the alignment community, this is a staggering result. For previous methods that have successfully demonstrated a high degree of field free 3D alignment (PRL 112, 173602), the alignment only persists for some fraction of a picosecond.

The result is also relevant – as pointed out by the authors – for potential studies of gas phase chemical and molecular dynamics. The authors show that larger molecules, perhaps of chemical interest, can also be aligned and will in fact remain aligned for longer periods of time. Measuring femtosecond time resolved dynamics in excited states of such systems in the molecular frame is an exciting prospect indeed. The authors also demonstrate the potential of alignment angle-resolved strong field physics experiments on larger systems by measuring the alignment dependence of strong field ionization of DBT molecules. This is a forward-looking demonstration, as I do not believe that this is currently calculable to any reasonable accuracy. It has emerged nonetheless that such measurements are needed –perhaps to start with on smaller systems (arXiv:1611.06476) – as quantitative benchmarks for theories of strong field physics (J. Phys. Chem. C, 2018, 122 (25), pp 13751 ; PRA 96, 043402).

Given the importance of this result for the alignment community, and its potential applications in strong field physics and molecular and chemical dynamics, I recommend this paper for publication in Nature Communications. However, some clarification is needed from the authors regarding each of these points, as detailed below.

Alignment

The authors have presented evidence that the long-time alignment behavior scales favorably to larger molecules. Nonetheless, for the asymmetric top molecules used in the study, the polarizability tensor is nearly aligned with the moment of inertia tensor. It would be useful to include a brief discussion or comment on the viability of 3D alignment when this is not the case. Particular examples would include axially chiral molecules – even small ones (H₂O₂) – or simply molecules with very low symmetry – propylene oxide for instance. These will have off-diagonal elements in either the polarizability or the moment of inertia, depending on the choice of molecular frame axes.

Further, in the case of TBI, it is not clearly to me if the MPA is also the fastest axis of rotation, or what the degree of anisotropy is for the polarizability tensor. It would be useful to mark the fastest and slowest axes of rotation for both DBT and TBI. Do these deviate significantly from the field free principal inertial axes? It also seems that only near-prolate tops were tested. Is it expected that the results would be much the same for near-oblate tops? This is not clear to me.

In general a brief discussion of the expectations for molecules with varying symmetry, and therefore varying polarizability and moment of inertia tensors would be valuable to readers in the alignment community. This is often overlooked in papers on alignment.

Molecular Dynamics

The authors have suggested, and it is indeed the case, that 3D aligned molecules under field free conditions will facilitate the study of excited state molecular dynamics in the molecular frame. While this is an exciting scientific prospect, it is important to note that the molecules in this study are 'dressed' by the He droplet. The authors should therefore briefly discuss the potential effects of this on excited state dynamics. Is the electronic structure of a molecule significantly effected? It is clear that the rotational dynamics are slowed significantly, is there a related effect on the vibrational structure? It is not immediately evident how closely the dynamics would map to the case of an isolated molecule. Even if it the case that such coupling is expected to occur, understanding the nature of this coupling is also scientifically valuable as it would serve as a bridge between the behavior of the isolated molecule and larger coupled systems. Some coauthors of this manuscript have already initiated similar studies in the case of rotation by impulsive alignment of a molecule in a He droplet (PRL 118 203203).

The authors should comment on the potential coupling of the He droplet and the excited vibronic states of the trapped molecule. It is necessary to have some understanding of this when embarking on the study of excited state dynamics.

Strong Field Physics

The authors have demonstrated the potential for molecular frame studies of strong field physics. Here too molecule-He coupling may be an issue that needs tackling, and is worth discussing.

Further, the authors have measured the strong field ionization yield as a function of alignment of only one axis of DBT. Note that is in general a 2D function, which varies with rotation of the body about the MPA as well (J. Phys. Chem. C, 2018, 122 (25), pp 13751, PRA 96, 043402). There is an experimental method that can measure this 2D angular dependence (arXiv: 1611.06476), but it is certainly restricted to small polyatomics (likely not applicable to either DBT or TBI). It would be very useful from the point of view of strong field physics to be able make such measurements, and it would therefore be worth discussing the experimental effort that is necessary to do this here.

The authors should discuss the potential effect of molecule-He coupling on strong field physics experiments. The authors should also discuss the possible experimental extension of the angle dependent strong field ionization measurement on DBT to a fully alignment-angle resolved measurement (i.e., the yield as a function of all relevant alignment angles.)

Once the above comments have been addressed, I recommend publication of this manuscript in Nature Communications

Reviewer #2 (Remarks to the Author):

Referee report

The paper by Chatterley et al. presents results on non-adiabatic molecular alignment in He-nanodroplets. The standard techniques for molecular alignment are either long or short pulse. In the

long pulse scheme, molecules show a high degree of alignment during the pulse, forcing experiments on the aligned molecules to take place in a strong alignment field. For short pulse experiments, this can get fixed due to rotational revivals (for small molecules), however the degree of alignment is considerably smaller compared to the long pulse case.

The paper presents a new and interesting aspect to this. A long pulse is used to adiabatically align the molecules. A sudden turn-off leaves the molecules aligned at a relatively high $\langle \cos^2\theta \rangle$ for some time. For molecules in the gas phase (here iodine and 4,4'-diiodobiphenyl), the alignment decays slower than the laser intensity drop, but still on a picosecond timescale, its exact value depending on the size and shape of the molecule. For molecules in He-nanodroplets, the alignment decay is delayed to several ten picoseconds, thus opening a relatively long window for time-resolved experiments on highly aligned molecules with very low field background from the alignment pulse.

The molecular alignment is quantified using the Coulomb explosion technique. While for gas phase experiments, this can be accomplished in a highly quantitative way, it requires corrections in He-nanodroplets due to scattering with the surrounding atoms. In addition to the 1d alignment with linearly polarized pulses on I₂ and 4,4'-diiodobiphenyl, the authors demonstrate 3d alignment using elliptically polarized laser pulses on the much more complex molecules dibromoterthiophene and tetrabromoindigo. Bromine and sulfur ions are detected in different projections to deliver the 3d alignment parameters. Essentially, the authors find the same long lasting alignment as was discovered in the 1d case.

This is a valuable, new and high impact contribution to the field of ultrafast molecular dynamics and in my opinion highly suitable for the general readership of Nature Communications. The results are technically sound, the presentation is on a very high level and the view on future applications is suitable for a general readership. The balance of main paper and appendix is just right. I highly recommend this paper for publication in Nature Communications.

I suggest a few minor edits before publication:

- I would suggest to mention the alignment laser wavelength as well as the peak intensity earlier in the paper since the authors refer to 1% of the peak intensity before mentioning it
- The authors mention ref 12 for a description of the correction regarding the ion-He scattering. It would be good to just mention in one or two sentences how this is done
- At the first mention of the molecules, it would be great to refer to the respective figure inset
- For the definition of the angle α , one should refer to the figure in the text
- Figure 2, y-axis : using the symbol $\langle \cos^2\theta \rangle / \langle \cos^2\alpha \rangle$ is misleading, as the authors do not plot the ratio but rather choose one or the other depending on color. One could maybe use 'or' on the axis
- Conclusions: The author should state more clearly, how singlet fission, exciplex formation and bimolecular reactions benefit from monitoring the parent ions

Reviewer #3 (Remarks to the Author):

The manuscript "Long-lasting field-free alignment of large molecules inside helium nanodroplets" describes the field-free and persistent alignment of several molecular species in a helium droplet. The experimental results are novel and interesting and deserve publication in Nature Communications. The manuscript structure and presentation, however, requires major revisions.

Reading the manuscript, I was dismayed by the lack of abstract and introduction, was pleasantly

surprised by the experimental results, and was dismayed again by the overly general and almost vaporous discussion. I believe that the manuscript requires, and deserves, a rather thorough revision.

The manuscript abstract does not fulfill the role of an abstract at all and should be completely rewritten. The abstract in its current form is a rambling introduction containing background information and vague relevance claims. I suggest to review established guidelines for scientific manuscript structure, e.g., the AIP style manual (cf.: <https://goo.gl/BqyGYF>). I cite: "An abstract [...] should be a concise summary of the significant items in the paper, including the results and conclusions. In combination with the title it must be an adequate indicator of the content of the article. [It] should not contain literature citations [...]."

The manuscript lacks a proper and properly structured introduction of the research field and preceding work. I had to go through a number of cited papers to piece together the context for the presented work -- this is unacceptable for a manuscript addressing the broad audience of Nature Chemistry. Again, I suggest following the established scientific article structure, e.g., according to the AIP manual:

"(1) Make the precise subject of the paper clear early in the introduction. As soon as possible, inform the reader what the paper is about. Depending on what you expect your typical reader already knows on the subject, you may or may not find it necessary to include historical background...
(2) Indicate the scope of coverage of the subject. Somewhere in the introduction state the limits within which you treat the subject. This definition of scope may include such things as the ranges of parameters dealt with, any restrictions made upon the general subject covered by the paper, and whether the work is theoretical or experimental.
(3) State the purpose of the paper. Every legitimate scientific paper has a purpose that distinguishes it from other papers on the same general subject."

All technical terms that are not obvious to the non-specialized reader (i.e., a broader audience of spectroscopists and molecular physicists) should be clearly introduced. E.g., the meaning of θ and $\cos^2(\theta)$, 1-D and 3D alignment. (I personally think that the latter terms are misleading: both refer to alignment in 3D space. A linearly polarized beam aligns a single molecular axis; an elliptically polarized beam aligns two molecular axes.)

Due to the lack of introduction, it is much harder to determine which aspects of the work are novel. The authors should properly introduce adiabatic alignment, rapid truncation of adiabatic alignment, alignment in He droplets, 1D and 3D alignment... Currently those aspects are introduced in the description of Fig. 1 with scattered references to earlier work, making it ambiguous which statements refer to the literature and which refer to the data shown.

The authors claim that lower rotational frequencies in He droplets lead to longer alignment durations - and repeat that claim for larger molecules (that have higher inertial moments and therefore lower state frequencies). I believe they are mistaken and only the temperature is relevant: The rotational states are occupied according to the Boltzmann distribution and states of similar frequency are occupied at similar temperatures, irrespective of the environment. The rotational transition frequencies follow the same proportionality law inside or outside of He droplets. Therefore, states with similar frequencies interfere to create alignment and the loss of alignment (= loss of constructive interference between the states) occurs on the same time-scale.

There is a higher density of states inside the droplets, which translates into a longer revival period, but this is independent of the alignment / revival width.

So only the temperature determines the width of the alignment peak (or revivals) inside or outside He

droplets. All molecules should show a similar alignment duration, as long as the Boltzmann distribution contains a reasonable number of rotational states. Data in Fig. 1b, 2a, 2b fulfill this expectation. I2 shows a much shorter alignment duration. Maybe this is due to the molecular symmetry?

The "Illustration of how 3D alignment of DBT is characterized." in Fig. 2 became only clear to me after referring to the supporting materials. Maybe the figure caption should attempt a more cursory and simple description and leave a detailed description for the supporting materials.

The "proof-of-principle [...] prototype linear dichroism experiment" is introduced in a weird fashion. The cited literature hardly mentions the term linear dichroism and does not support the prototypical nature of the work. It is not explained how the alignment method would improve the experiments quoted as ref. [18-21]. It reads as if this section was tacked onto the manuscript without the will to write a meaningful introduction or discussion of the results.

The summary/discussion section of the manuscript feels cobbled-together. What do the authors want to learn from fs time-resolve imaging using Coulomb explosion, and how would alignment in He droplets help? What information would be gained from time-resolved imaging through linear dichroism, and could this address longstanding questions? How practical is a time-resolved diffraction experiment and would the achieved degree of alignment provide meaningful diffraction results that go significantly beyond that obtained for unaligned molecules? ... All of those questions arise from a single sentence and remain unanswered. I would be much more interested if the authors could describe and discuss one or more specific future experiments as opposed to razing the field in the broadest terms.

The results of the work should be properly discussed and placed into context. What methods of alignment were used in the past (maybe in context of the mentioned spectroscopy or diffraction experiments)? How much of an advantage offers the current method for those experiments? What is the relative importance of the better alignment versus longer alignment duration? What is the nature of the alignment at $t=0$ (average angle (phase) of zero \pm variance due to imperfect alignment) versus $t=20$ ps (average angle (phase) of $xyz \pm$ variance)? How would that affect the proposed spectroscopic experiments? There are a lot of interesting questions that could be discussed by the expert authors to inspire the next generations of spectroscopists.

With best regards,
Thomas Schultz

Reviewers' comments:

Reviewer #1 (Remarks to the Author):

In this manuscript the authors describe the three dimensional (3D) alignment of asymmetric top molecules in a helium (He) droplet under essentially laser-field free conditions. A combination of sophisticated experimental techniques are used to generate and measure the alignment. An elliptically polarized switched laser pulse adiabatically aligns axes of largest and intermediate polarizability of the molecules via rotational Raman scattering. As the pulse abruptly switches off, the molecules are free to rotate again and the alignment decays, precluding field free alignment. This problem was highlighted by Underwood et al. (PRL 97 173001), resulting in this method being previously abandoned as a route to field free three dimensional alignment. The authors have very clearly demonstrated here that in the case of a molecule trapped in a He droplet, not only is this method successful, but can also provide 3D alignment under field free conditions for tens of picoseconds as a result of coupling between the rotations of the molecule to its solvation shell within the droplet. From the point view of the alignment community, this is a staggering result. For previous methods that have successfully demonstrated a high degree of field free 3D alignment (PRL 112, 173602), the alignment only persists for some fraction of a picosecond.

The result is also relevant – as pointed out by the authors – for potential studies of gas phase chemical and molecular dynamics. The authors show that larger molecules, perhaps of chemical interest, can also be aligned and will in fact remain aligned for longer periods of time. Measuring femtosecond time resolved dynamics in excited states of such systems in the molecular frame is an exciting prospect indeed. The authors also demonstrate the potential of alignment angle-resolved strong field physics experiments on larger systems by measuring the alignment dependence of strong field ionization of DBT molecules. This is a forward-looking demonstration, as I do not believe that this is currently calculable to any reasonable accuracy. It has emerged nonetheless that such measurements are needed –perhaps to start with on smaller systems (arXiv:1611.06476) – as quantitative benchmarks for theories of strong field physics (J. Phys. Chem. C, 2018, 122 (25), pp 13751 ; PRA 96,043402).

Given the importance of this result for the alignment community, and its potential applications in strong field physics and molecular and chemical dynamics, I recommend this paper for publication in Nature Communications. However, some clarification is needed from the authors regarding each of these points, as detailed below.

We thank referee 1 for the careful reading of our manuscript and the positive comments. We are very pleased they share our enthusiasm for this novel technique.

Alignment

The authors have presented evidence that the long-time alignment behavior scales favorably to larger molecules. Nonetheless, for the asymmetric top molecules used in the study, the polarizability tensor is nearly aligned with the moment of inertia tensor. It would be useful to include a brief discussion or comment on the viability of 3D alignment when this is not the case. Particular examples would include axially chiral molecules – even small ones (H₂O₂) – or simply molecules with very low symmetry – propylene oxide for instance. These will have off-diagonal elements in either the polarizability or the moment of inertia, depending on the choice of molecular frame axes.

Further, in the case of TBI, it is not clearly to me if the MPA is also the fastest axis of rotation, or what the degree of anisotropy is for the polarizability tensor. It would be useful to mark the fastest and slowest axes of rotation for both DBT and TBI. Do these deviate significantly from the field free principal inertial axes? It also seems that only near-prolate tops were tested. Is it expected that the results would be much the same for near-oblate tops? This is not clear to me.

In general a brief discussion of the expectations for molecules with varying symmetry, and therefore varying polarizability and moment of inertia tensors would be valuable to readers in the alignment community. This is often overlooked in papers on alignment.

This is a very important point. We have now included the polarizability tensors for all molecules studied in the supplementary information. The polarizability anisotropies are rather large, especially for TBI. The reviewer is correct in saying that in the polarizability frame, the moment of inertia tensor for TBI and DBT is non-diagonal. However, we do not believe that symmetry considerations will considerably affect our method. As we are not dependent on rotational revivals, the spacing of rotational levels (which the symmetry affects) is unlikely to play a major role. Any effect of symmetry is likely much more subtle than the slow dynamics observed in our experiments, and would require careful (and extremely difficult) theory to fully deduce. We have now included a sentence in the manuscript mentioning our insensitivity to symmetry.

Molecular Dynamics

The authors have suggested, and it is indeed the case, that 3D aligned molecules under field free conditions will facilitate the study of excited state molecular dynamics in the molecular frame. While this is an exciting scientific prospect, it is important to note that the molecules in this study are ‘dressed’ by the He droplet. The authors should therefore briefly discuss the potential effects of this on excited state dynamics. Is the electronic structure of a molecule significantly effected? It is clear that the rotational dynamics are slowed significantly, is there a related effect on the vibrational structure? It is not immediately evident how closely the dynamics would map to the case of an isolated molecule. Even if it the case that such coupling is expected to occur, understanding the nature of this coupling is also scientifically valuable as it would serve as a bridge between the

behavior of the isolated molecule and larger coupled systems. Some coauthors of this manuscript have already initiated similar studies in the case of rotation by impulsive alignment of a molecule in a He droplet (PRL 118 203203).

The authors should comment on the potential coupling of the He droplet and the excited vibronic states of the trapped molecule. It is necessary to have some understanding of this when embarking on the study of excited state dynamics.

The referee is quite right that all experiments performed using this scheme will necessarily be on helium dressed molecules. For many dynamics experiments, we anticipate that the effect of the helium solvent will be rather minimal: high resolution electronic and vibrational spectroscopy show that the perturbation is very small (<https://doi.org/10.1002/anie.200300611>), so we anticipate that the dynamics will not change significantly. In cases where the perturbation is significant, as the referee points out, this provides an opportunity to study basic solvation effects.

We have added a paragraph to the end of the discussion section commenting on this.

Strong Field Physics

The authors have demonstrated the potential for molecular frame studies of strong field physics. Here too molecule-He coupling may be an issue that needs tackling, and is worth discussing.

Further, the authors have measured the strong field ionization yield as a function of alignment of only one axis of DBT. Note that is in general a 2D function, which varies with rotation of the body about the MPA as well (J. Phys. Chem. C, 2018, 122 (25), pp 13751, PRA 96, 043402). There is an experimental method that can measure this 2D angular dependence (arXiv:1611.06476), but it is certainly restricted to small polyatomics (likely not applicable to either DBT or TBI). It would be very useful from the point of view of strong field physics to be able make such measurements, and it would therefore be worth discussing the experimental effort that is necessary to do this here.

The authors should discuss the potential effect of molecule-He coupling on strong field physics experiments. The authors should also discuss the possible experimental extension of the angle dependent strong field ionization measurement on DBT to a fully alignment-angle resolved measurement (i.e., the yield as a function of all relevant alignment angles.)

The section on alignment-dependent ionization of DBT was originally included as a simple proof-of-principle, however the referee is correct that it points the way to many possible strong-field experiments in helium droplets. As DBT is a large, asymmetric, flexible molecule, accurate theoretical modelling of it would be exceedingly challenging. However, for these reasons it is an excellent example of a molecule that could not be studied in the molecular frame using other techniques. For detailed molecular strong-field experiments in helium, we would certainly want to start with simpler molecular systems.

As to the question of resolving the full 2D ionization yield function, this is certainly not easy. Our 3D alignment technique confines the molecule to a plane perpendicular to the laser propagation, and only

the angle of the molecule inside this plane can be freely varied. In theory, one could use two mutually perpendicular alignment beams to gain full control of both the yaw and pitch, and truncate both pulses simultaneously to produce field-free alignment. Such an experiment is, however, at the absolute edge of feasibility with current technology.

We have now included a paragraph in the discussion section that discusses the possibilities for strong field experiments inside helium that this technique opens up. We also briefly discuss the opportunities and difficulties associated with mapping the full 2D ion yield function.

Once the above comments have been addressed, I recommend publication of this manuscript in Nature Communications

We thank referee 1 again for their careful reading and constructive remarks.

Reviewer #2 (Remarks to the Author):

Referee report

The paper by Chatterley et al. presents results on non-adiabatic molecular alignment in He-nanodroplets. The standard techniques for molecular alignment are either long or short pulse. In the long pulse scheme, molecules show a high degree of alignment during the pulse, forcing experiments on the aligned molecules to take place in a strong alignment field. For short pulse experiments, this can get fixed due to rotational revivals (for small molecules), however the degree of alignment is considerably smaller compared to the long pulse case.

The paper presents a new and interesting aspect to this. A long pulse is used to adiabatically align the molecules. A sudden turn-off leaves the molecules aligned at a relatively high $\langle \cos^2\theta \rangle$ for some time. For molecules in the gas phase (here iodine and 4,4'-diiodobiphenyl), the alignment decays slower than the laser intensity drop, but still on a picosecond timescale, its exact value depending on the size and shape of the molecule. For molecules in He-nanodroplets, the alignment decay is delayed to several ten picoseconds, thus opening a relatively long window for time-resolved experiments on highly aligned molecules with very low field background from the alignment pulse. The molecular alignment is quantified using the Coulomb explosion technique. While for gas phase experiments, this can be accomplished in a highly quantitative way, it requires corrections in He-nanodroplets due to scattering with the surrounding atoms. In addition to the 1d alignment with linearly polarized pulses on I₂ and 4,4'-diiodobiphenyl, the authors demonstrate 3d alignment using elliptically polarized laser pulses on the much more complex molecules dibromoterthiophene and tetrabromoindigo. Bromine and sulfur ions are detected in different projections to deliver the 3d alignment parameters. Essentially, the authors find the same long lasting alignment as was discovered in the 1d case.

This is a valuable, new and high impact contribution to the field of ultrafast molecular dynamics and in my opinion highly suitable for the general readership of Nature Communications. The results are technically sound, the presentation is on a very high level and the view on future applications is suitable for a general readership. The balance of main paper and appendix is just right. I highly recommend this paper for publication in Nature Communications.

Many thanks to this referee for their careful reading. We are delighted that they share our enthusiasm for this new technique.

I suggest a few minor edits before publication:

- I would suggest to mention the alignment laser wavelength as well as the peak intensity earlier in the paper since the authors refer to 1% of the peak intensity before mentioning it
- The authors mention ref 12 for a description of the correction regarding the ion-He scattering. It would be good to just mention in one or two sentences how this is done
- At the first mention of the molecules, it would be great to refer to the respective figure inset
- For the definition of the angle α , one should refer to the figure in the text
- Figure 2, y-axis : using the symbol $\frac{\cos^2\theta}{\cos^2\alpha}$ is misleading, as the authors do not plot the ratio but rather choose one or the other depending on color. One could maybe use 'or' on the axis

All of these suggestions have now been implemented.

- Conclusions: The author should state more clearly, how singlet fission, exciplex formation and bimolecular reactions benefit from monitoring the parent ions

We have significantly reworked the conclusions section of the paper, to better discuss the future experimental possibilities our technique opens up. The referee is correct that we were somewhat vague before.

Reviewer #3 (Remarks to the Author):

This referee also included comments marked up on a PDF. We address those comments that are not reflected here at the end of our response. Minor points in the marked up PDF have been corrected without further comment.

The manuscript "Long-lasting field-free alignment of large molecules inside helium nanodroplets" describes the field-free and persistent alignment of several molecular species in a helium droplet. The experimental results are novel and interesting and deserve publication in Nature Communications. The manuscript structure and presentation, however, requires major revisions.

Reading the manuscript, I was dismayed by the lack of abstract and introduction, was pleasantly surprised by the experimental results, and was dismayed again by the overly general and almost vaporous discussion. I believe that the manuscript requires, and deserves, a rather thorough revision.

The manuscript abstract does not fulfill the role of an abstract at all and should be completely rewritten. The abstract in its current form is a rambling introduction containing background information and vague relevance claims. I suggest to review established guidelines for scientific manuscript structure, e.g., the AIP style manual (cf.: <https://goo.gl/BqyGYF>). I cite: "An abstract [...] should be a concise summary of the significant items in the paper, including the results and conclusions. In combination with the title it must be an adequate indicator of the content of the article. [It] should not contain literature citations [...]."

The manuscript lacks a proper and properly structured introduction of the research field and preceding work. I had to go through a number of cited papers to piece together the context for the presented work -- this is unacceptable for a manuscript addressing the broad audience of Nature Chemistry. Again, I suggest following the established scientific article structure, e.g., according to the AIP manual:

"(1) Make the precise subject of the paper clear early in the introduction. As soon as possible, inform the reader what the paper is about. Depending on what you expect your typical reader already knows on the subject, you may or may not find it necessary to include historical background...

(2) Indicate the scope of coverage of the subject. Somewhere in the introduction state the limits within which you treat the subject. This definition of scope may include such things as the ranges of parameters dealt with, any restrictions made upon the general subject covered by the paper, and whether the work is theoretical or experimental.

(3) State the purpose of the paper. Every legitimate scientific paper has a purpose that distinguishes it from other papers on the same general subject."

[Redacted] The paper was originally in this format.

Now we have been able to revise the paper into the Nature Communications style, which in particular means that we have been able to include a quite detailed introduction. The abstract is rather close to our original abstract [Redacted] The referee is quite right that in its submitted form, the paper was rather unusual compared to most of the scientific literature (although complying with the Nature Physics format).

All technical terms that are not obvious to the non-specialized reader (i.e., a broader audience of spectroscopists and molecular physicists) should be clearly introduced. E.g., the meaning of θ and $\cos^2(\theta)$, 1-D and 3D alignment. (I personally think that the latter terms are misleading: both refer to alignment in 3D space. A linearly polarized beam aligns a single molecular axis; an elliptically polarized beam aligns two molecular axes.)

We now include a more detailed description of the differences between 1D and 3D alignment in the introduction. Confusing as these terms may be, they are the ones that the community has landed on, so using any other terminology would simply create more confusion. $\langle \cos^2\theta_{2D} \rangle$ is defined in the first paragraph of the results section.

Due to the lack of introduction, it is much harder to determine which aspects of the work are novel. The authors should properly introduce adiabatic alignment, rapid truncation of adiabatic alignment, alignment in He droplets, 1D and 3D alignment... Currently those aspects are introduced in the description of Fig. 1 with scattered references to earlier work, making it ambiguous which statements refer to the literature and which refer to the data shown.

We have now included a much more detailed introduction, in the Nature Communications style.

The authors claim that lower rotational frequencies in He droplets lead to longer alignment durations -- and repeat that claim for larger molecules (that have higher inertial moments and therefore lower state frequencies). I believe they are mistaken and only the temperature is relevant: The rotational states are occupied according to the Boltzmann distribution and states of similar frequency are occupied at similar temperatures, irrespective of the environment. The rotational transition frequencies follow the same proportionality law inside or outside of He droplets. Therefore, states with similar frequencies interfere to create alignment and the loss of alignment (= loss of constructive interference between the states) occurs on the same time-scale. There is a higher density of states inside the droplets, which translates into a longer revival period, but this is independent of the alignment / revival width. So only the temperature determines the width of the alignment peak (or revivals) inside or outside He droplets. All molecules should show a similar alignment duration, as long as the Boltzmann distribution contains a reasonable number of rotational states. Data in Fig. 1b, 2a, 2b fulfill this expectation. I₂ shows a much shorter alignment duration. Maybe this is due to the molecular symmetry?

The referee is quite right that, in the gas phase, the time for loss of alignment is a function solely of the moments of inertia and the temperature of the molecules. Molecules in helium nanodroplets have a low (0.4 K) temperature, and increased effective moments of inertia relative to the gas phase (<https://doi.org/10.1002/anie.200300611>), so it is natural to question as to whether these effects can explain the slow decrease in alignment after truncation.

To test whether this could be the case, we performed a number of numerical simulations on gas phase I₂, now given in the supplementary material. We simulated the rotational dynamics of I₂ after pulse truncation, with differing rotational temperatures and effective moments of inertia.

We found that temperature effects alone definitely cannot explain the slow drop of alignment, as at 0.4 K, gaseous I₂ still loses all alignment in <20 ps. Likewise, increased effective moment of inertia can be ruled out, as 10-fold increase is required to match the experimental drop time. Such a large increased moment of inertia is unphysical – the best estimate of the true increase is ×1.7 (<https://doi.org/10.1103/PhysRevLett.118.203203>). Overall, we conclude that further effects of the helium solvent prevent the molecules from rapidly losing their alignment. This echoes previous findings from our group, that alignment dynamics in helium droplets cannot be explained as perturbed isolated dynamics (<https://doi.org/10.1103/PhysRevLett.118.203203>).

The "Illustration of how 3D alignment of DBT is characterized." in Fig. 2 became only clear to me after referring to the supporting materials. Maybe the figure caption should attempt a more cursory and simple description and leave a detailed description for the supporting materials.

We have now revised this description in the text somewhat to be clearer

The "proof-of-principle [...] prototype linear dichroism experiment" is introduced in a weird fashion. The cited literature hardly mentions the term linear dichroism and does not support the prototypical nature of the work. It is not explained how the alignment method would improve the experiments quoted as ref. [18-21]. It reads as if this section was tacked onto the manuscript without the will to write a meaningful introduction or discussion of the results.

We now have the space to treat this section a little more thoroughly. We now refer to this experiment as a 'molecular frame strong-field ionization' experiment, as this fits better with the existing literature (although it certainly is also a linear dichroism experiment). We have added an explanation of the advantages of our method as compared to previous demonstrations, namely that there are no symmetry requirements of the molecule, and intact cations are retrieved.

We now also include a discussion of strong field experiments in helium droplets at the end of the paper. We briefly comment on the challenges and opportunities of molecular frame strong-field physics in helium droplets, as well as discussing extensions to higher dimensioned experiments as referee 2 suggested.

The summary/discussion section of the manuscript feels cobbled-together. What do the authors want to learn from fs time-resolved imaging using Coulomb explosion, and how would alignment in He droplets help? What information would be gained from time-resolved imaging through linear dichroism, and could this address longstanding questions? How practical is a time-resolved diffraction experiment and would the achieved degree of alignment provide meaningful diffraction results that go significantly beyond that obtained for unaligned molecules? ... All of those questions arise from a single sentence and remain unanswered. I would be much more interested if the authors could describe and discuss one or more specific future experiments as opposed to raising the field in the broadest terms.

The results of the work should be properly discussed and placed into context. What methods of alignment were used in the past (maybe in context of the mentioned spectroscopy or diffraction experiments)? How much of an advantage offers the current method for those experiments? What is the relative importance of the better alignment versus longer alignment duration? What is the nature of the alignment at $t=0$ (average angle (phase) of zero \pm variance due to imperfect alignment) versus $t=20$ ps (average angle (phase) of xyz \pm variance)? How would that affect the proposed spectroscopic experiments? There are a lot of interesting questions that could be discussed by the expert authors to inspire the next generations of spectroscopists.

The new discussion section of the paper addresses these issues. We discuss the levels of alignment required for two different experiments, molecular frame photoelectron angular distribution measurements, and ultrafast X-ray diffraction. For the former, we have at least a 4 ps window to perform experiments on DIBP, while the latter requires higher alignment and some experimental modifications before we can exploit a long field-free window. We now discuss what steps would be required to gain 10 ps or more field-free aligned windows for experiments like these.

As mentioned above, we also now discuss the experimental limitations and opportunities provided by the helium environment these experiments must be performed in. Briefly, we anticipate that the helium minimally perturbs dynamics, and where it does there are exciting opportunities to study solvation effects. Helium also offers unique opportunities to build non-covalent complexes. We have recently begun to use our field-free alignment technique to determine structures of weak complexes of polyaromatic hydrocarbon systems.

Comments from referee 3 in the marked up PDF:

P2, Re: rotational revival lifetimes - The time-scale of rotational motion and resulting loss of alignment is a function of temperature (occupied states) and an inverse function of molecular size (growth of inertial moment with size). There is no inherent property that would restrict alignment to times < 1 ps.

Can you cite literature to back up the claim that field-free alignment is currently only possible for small linear molecules and for times < 1 ps? Or maybe rephrase the statement to say that past experimental implementations only obtained alignment for < 1 ps and for small linear molecules.

The referee is correct that in principle it is possible to create rotational revivals lasting > 1 ps, if the molecule is very large and the temperature very low ($\ll 1$ K). However, large molecules cannot generally be cooled to these very low temperatures, so we know of no feasible way of creating the conditions required for long-lived revivals. In any case, we have modified the text to include the word 'typically'.

P3, Re: rotational dephasing - I think the word 'dephasing' should be replaced with 'loss of alignment'. Dephasing of the coherent rotational wavepacket in isolated molecules occurs on a much longer time scale, as is evident from the observed revivals described in ref. [8]. I think the field of NMR managed to mess up our use of 'dephasing', because they use it in the description of heterogeneous effects. I would use 'dephasing' to describe the loss of phase information in an ensemble of quantum states and not just the loss of constructive interference.

I am also not sure, how dephasing is characterizing free molecules.

The referee is quite right, 'dephasing' is not the precise term to use for gas phase molecules, as there is nothing removing the phase relationship. We have changed this to 'dispersion', which more accurately reflects the process, and certainly characterizes free molecules.

P4, Re: 'fortuitous' effects - This point belongs into the discussion, together with the discussion of alignment (Raman) cross sections and efficiencies as function of molecular size and shape.

I believe that this statement is wrong (see last comment on p. 3): larger molecules have a higher density of states but the Boltzmann distribution will lead to a similar frequency distribution of occupied states.

If this statement is true, it is not 'fortuitous', but simply science.

As discussed above, we believe there is more to the alignment dynamics inside helium droplets than just temperature and density of states effects. As the slow dynamics are due to interactions between the molecule and the helium, it is reasonable that larger molecules will be impeded more. We have, however, removed the word 'fortuitous'. During drafting, there was intense debate amongst the authors as to whether this word was appropriate, and we will give the referee the deciding vote.

P5, Re: 'previous works' - Please give citations for previous works measuring the alignment of DBT. Or maybe you want to say that the alignment is smaller than that observed for DIBP?

This has now been corrected to be clear that we were comparing to our previous work where we measured 3D alignment of dichloriodobenzene.

P9, Re: filtering ion kinetic energy - How much does this signal filtering (selection of particular ion kinetic energies) affect the results? You suggest that your method will be useful for other spectroscopy or diffraction experiments, but those cannot use similar filtering. Would this affect the results and limit the usefulness of your alignment method?

We have now expanded the methods section. The filtering method does not actually select molecules that were well aligned. Instead, it selects the events, which have lower levels of non-axial recoil, and so are more representative of the true alignment distribution. We have now expanded this section in the text, and included a reference.

Reviewers' comments:

Reviewer #1 (Remarks to the Author):

After reading the new version of the manuscript, I find that the authors have satisfactorily addressed my comments. After making a minor addition, I recommend that the manuscript be published in Nature Communications.

In addition to the polarizability tensor, now clearly defined for all the molecules in the supplement, the authors should also mark the a, b and c axes of rotation. If possible, including the inertia tensor for each case would also be useful to readers in the alignment community.

Reviewer #3 (Remarks to the Author):

The revised manuscript has greatly gained in clarity. I supply an edited manuscript with suggested modifications to further clarify the text, in particular in the introduction section. The abstract still seems to violate most rules for scientific writing, but I leave it to the editor to determine whether this abstract style is suited for Nat. Comm..

I still object to several imprecise and possibly misleading statements that should not be published as written. The authors greatly expanded the number of cited papers, but failed to provide references to clarify some issues that -- to me -- seem questionable.

The authors still state in the abstract that "[Field-free alignment] is only possible for small, linear molecules and for times less than 1 picosecond." Later in the manuscript, they modified their statements to say this is "typically" the case (see P2 in the rebuttal letter). Let me point to the oldest experimental papers on rotational alignment, e.g. [Heritage, J. P., et al. (1975). "Observation of Coherent Transient Birefringence in Cs₂ Vapor." Physical Review Letters 34(21): 1299-1302.], which showed molecular alignment of CS₂ for 10s of picoseconds -- as mentioned in the first round and explained in more detail below, the duration of alignment (or revivals) is purely a function of the excitation process or, if broad-band excitation is used, a function of the molecular temperature. I cannot assess the I2 simulation the authors mention in the supplementary information, but a simulation does not in any way change the fundamental physics of alignment, which the authors should think through before publishing their data.

The authors should clarify vague statements about the "impeding effect" of the helium environment. If the authors have a model in mind of how the helium impedes the rotational motion, then this would surely be the central insight gained from their work! And it surely deserve a proper discussion in the discussion section and should not be dropped a factual statement in the introduction. The hindrance might be imagined as either librational motion in the residual laser field, which would weaken the claim of 'field free alignment', or as rotational cooling due to angular momentum transfer to the droplet on a timescale faster than the rotational dispersion. The latter implies loss of coherence and would be easily proven by scanning longer delay times to observe the presence or absence of wave packet modulation. I cannot comment on whether rotational cooling is plausible on a picosecond time scale, the data in the authors independent manuscript on "Laser-induced rotation of iodine molecules in He-nanodroplets..." (<https://arxiv.org/pdf/1702.01977.pdf>) indicates long-lasting coherence and the coherence lifetime does not depend on the excitation type (impulsive excitation in the mentioned manuscript).

Let me now make a number of statements concerning the physics of alignment, which I hope will clarify why I do not like the hand-waving discussion of an impeding helium environment.

The timescale for revival lifetimes and the loss of alignment in helium droplets is governed by the same quantum physical laws that describe molecular rotation. If you consider rotation in He to be classical, then please consider this as an extension of quantized rotation towards a large density of rotational states. These laws are well understood and the manuscript should avoid any statements that imply mysterious effects in the helium droplets.

Any field-free aligned state must be described as the coherent sum of rotational states. This state superposition can be described in the time- or the frequency-domain, but those two descriptions are equivalent and interchangeable. The Fourier transformation allows us to convert between frequency (spectrum) and time domain (wavepacket dynamics) representation without any loss of information.

The quantum chemical description of rotational states (the spectrum) is well established. The role of a helium environment on rotational states has been investigated for small molecules and seems to reduce the effective rotational constants by a factor 1.01-4.5 (<https://doi.org/10.1002/anie.200300611>, as mentioned by the authors) or even close to 6 (for n_2O , see: <https://arxiv.org/pdf/1703.06753.pdf>). The authors themselves published a manuscript implying that the lowest rotational states of I_2 in helium might have a greatly enlarged moment of inertia due to a fixed first solvation shell ("Laser-induced rotation of iodine molecules in He-nanodroplets..." <https://arxiv.org/pdf/1702.01977.pdf>), which would translate into a much smaller effective rotational constant. The long-lasting temporal signal modulations in the data shown therein indicate that the observed wavepacket dynamics are still due to discrete quantized states that show coherence over nanosecond periods; i.e., there seems to be a discrete quantized spectrum. Analyzing the spectrum of such low-lying states might be interesting in itself.

Based on the above-mentioned Fourier relation between time- and frequency properties of the wave packet, we can make some unambiguous statements about the possible degree and duration of alignment:

(1) The duration of the alignment depends on the frequency differences between the involved states. A broad spectrum of state frequencies will always cause a short duration of alignment. A narrow ('cold') spectrum will cause a long alignment duration. Excitation can lead to a narrow spectrum either because the sample is cold (e.g., 0.4 K in He) and transitions selection rules limit the energy introduced by the excitation step (e.g., $\Delta J = \pm 2$ Raman selection rules), or because a narrow bandwidth excitation source is used (cf. the 1975 Heritage paper).

(2) The maximum degree of alignment depends on the number of involved states and also on their spectrum. We should think about alignment as destructive interference of transition terms from all wave packet states into the final detected state. Superposition of the states can lead to destructive interference along some spatial directions. This clarifies a theoretical limit for alignment: A higher degree of alignment requires the superposition of more quantum states. There is a direct trade-off between the number of states involved and the obtainable degree of alignment. Heritage in 1975 had lousy, but long-lasting alignment.

(3) When aiming for long-lasting and strong alignment, points (1) and (2) indicate that we must find a compromise between (a) long lasting, or (b) strong alignment. Point (a) requires a wave packet of only the lowest frequency states for long-lasting alignment, but the alignment will be not very strong. Point (b) requires a wave packet involving many states, but the duration of the alignment is now reduced because the states span a larger frequency range.

From the theoretical viewpoint laid out above, a long-lasting and strong alignment is only possible if there are many states (-> strong alignment allowed) with very low frequencies (-> long-lasting alignment is possible). The rotational frequency reduction in helium droplets may therefore allow fundamentally better and longer-lasting alignment than possible for gas-phase species, but the improvement is directly proportional to the reduction in rotational frequencies. Note that larger molecules have inherently lower rotational frequencies and can be better aligned for longer times (at least theoretically).

To summarize, it is important to remember that the degree and duration of alignment will always directly reflect the involved quantum states (even if the experimental realization is far from the theoretical limit) and it is useful to think about the frequency spectrum and the temperature of the molecular ensemble.

A final comment on the authors modeling of I2 alignment (supp. info.). I have no experience with modeling truncated adiabatic alignment, but if I assume that the excitation somehow resembles a Raman excitation at a 0.4 K equilibrated temperature ($\Delta J = \pm 2$ transitions from the states occupied with Boltzmann equilibrium) then I would easily expect such long-lived alignment. 0.4 K corresponds to an energy of $kT/h = 8.3$ GHz, i.e., a state with energy 8 GHz has $1/e$ population (*degeneracy) compared to the lowest rotational state. That would leave the $J=0$ and $J=2$ states as predominantly populated states for I2, with Raman transitions up to $J=4$; difference frequencies < 20 GHz and corresponding revival periods > 50 ps. I must assume that the proper modeling of the excitation process by the authors somehow leads to population of higher rotational states (i.e., involving multiple Raman transitions), or a non-equilibrium population of rotational states? This model then clearly disagrees with the observed long rotational dispersion time and cannot teach us anything about the observed systems - but it also does not detract from our knowledge about the required superposition of rotational states that would produce alignment signals such as observed.

Reviewer #1 (Remarks to the Author):

After reading the new version of the manuscript, I find that the authors have satisfactorily addressed my comments. After making a minor addition, I recommend that the manuscript be published in Nature Communications.

Great!

In addition to the polarizability tensor, now clearly defined for all the molecules in the supplement, the authors should also mark the a, b and c axes of rotation. If possible, including the inertia tensor for each case would also be useful to readers in the alignment community.

We have expanded Table 1 in the supplementary material to include the inertia tensor for each of the four molecules treated in the manuscript.

Reviewer #3 (Remarks to the Author):

The revised manuscript has greatly gained in clarity. I supply an edited manuscript with suggested modifications to further clarify the text, in particular in the introduction section. The abstract still seems to violate most rules for scientific writing, but I leave it to the editor to determine whether this abstract style is suited for Nat. Comm..

We are glad to hear that the manuscript is more clear now.

I still object to several imprecise and possibly misleading statements that should not be published as written. The authors greatly expanded the number of cited papers, but failed to provide references to clarify some issues that -- to me -- seem questionable.

The authors still state in the abstract that "[Field-free alignment] is only possible for small, linear molecules and for times less than 1 picosecond." Later in the manuscript, they modified their statements to say this is "typically" the case (see P2 in the rebuttal letter). Let me point to the oldest experimental papers on rotational alignment, e.g. [Heritage, J. P., et al. (1975). "Observation of Coherent Transient Birefringence in Cs₂ Vapor." Physical Review Letters 34(21): 1299-1302.], which showed molecular alignment of CS₂ for 10s of picoseconds -- as mentioned in the first round and explained in more detail below, the duration of alignment (or revivals) is purely a function of the excitation process or, if broad-band excitation is used, a function of the molecular temperature. I cannot assess the I2 simulation the authors mention in the supplementary information, but a simulation does not in any way change the fundamental physics of alignment, which the authors should think through before publishing their data.

Actually, the alignment observed by Heritage et al. in the 1975 PRL only last a few ps NOT 10s of ps (please note that the y-axis of Fig. 1 in the paper is on a logarithmic scale). Furthermore, the

degree of alignment is very, very low due to the low intensity of the alignment pulse (based on their experimental parameters we calculated, by solving the Schr. Equation, that the max $\langle \cos^2 \theta \rangle = 0.33 + 0.0000065$; note 0.33 is the value for randomly oriented molecules). By contrast, our work focuses on strong alignment needed for applications in molecular science. In this regime we are not aware of any field-free alignment lasting for more than ~ 1 ps (actually, for the rotational wave arguments given by the referee below).

To further emphasize that our work concerns strong alignment, we have added 'strong' in front of 'alignment' in line 4 of the abstract.

The authors should clarify vague statements about the "impeding effect" of the helium environment. If the authors have a model in mind of how the helium impedes the rotational motion, then this would surely be the central insight gained from their work! And it surely deserve a proper discussion in the discussion section and should not be dropped a factual statement in the introduction. The hindrance might be imagined as either librational motion in the residual laser field, which would weaken the claim of 'field free alignment', or as rotational cooling due to angular momentum transfer to the droplet on a timescale faster than the rotational dispersion. The latter implies loss of coherence and would be easily proven by scanning longer delay times to observe the presence or absence of wave packet modulation. I cannot comment on whether rotational cooling is plausible on a picosecond time scale, the data in the authors independent manuscript on "Laser-induced rotation of iodine molecules in He-nanodroplets..." (<https://arxiv.org/pdf/1702.01977.pdf>) indicates long-lasting coherence and the coherence lifetime does not depend on the excitation type (impulsive excitation in the mentioned manuscript).

We agree that the statement of an "impeding effect" on page 3, line 10 was not well documented and that the two references previously given on page 4, line -13, insufficient because the notion that helium impedes the molecular rotation is not explicitly stated in any of those two references. Therefore, we changed the sentence on page 3, line 10 to:

". . . thanks to the impeding effect of the He environment on molecular rotation observed earlier [17,20,21].

The two new references x,y, are those where we have previously explicitly stated that the helium environment impedes the rotational motion of the molecules. Reference z is maintained, because this is the first work where we observed a slowing of the rotational dynamics and discussed its possible origin.

The same three references are now also used on page 4, line -13:

". . . By contrast, in the droplets the impeding effect of the He environment on the molecular rotation [17,20,21] . . ."

Let me now make a number of statements concerning the physics of alignment, which I hope will clarify why I do not like the hand-waving discussion of an impeding helium environment.

The timescale for revival lifetimes and the loss of alignment in helium droplets is governed by the

same quantum physical laws that describe molecular rotation. If you consider rotation in He to be classical, then please consider this as an extension of quantized rotation towards a large density of rotational states. These laws are well understood and the manuscript should avoid any statements that imply mysterious effects in the helium droplets.

Any field-free aligned state must be described as the coherent sum of rotational states. This state superposition can be described in the time- or the frequency-domain, but those two descriptions are equivalent and interchangeable. The Fourier transformation allows us to convert between frequency (spectrum) and time domain (wavepacket dynamics) representation without any loss of information.

The quantum chemical description of rotational states (the spectrum) is well established. The role of a helium environment on rotational states has been investigated for small molecules and seems to reduce the effective rotational constants by a factor 1.01-4.5

(<https://doi.org/10.1002/anie.200300611>, as mentioned by the authors) or even close to 6 (for n_2O , see: <https://arxiv.org/pdf/1703.06753.pdf>). The authors themselves published a manuscript implying that the lowest rotational states of I_2 in helium might have a greatly enlarged moment of inertia due to a fixed first solvation shell ("Laser-induced rotation of iodine molecules in He-nanodroplets..." <https://arxiv.org/pdf/1702.01977.pdf>), which would translate into a much smaller effective rotational constant. The long-lasting temporal signal modulations in the data shown therein indicate that the observed wavepacket dynamics are still due to discrete quantized states that show coherence over nanosecond periods; i.e., there seems to be a discrete quantized spectrum. Analyzing the spectrum of such low-lying states might be interesting in itself.

Based on the above-mentioned Fourier relation between time- and frequency properties of the wave packet, we can make some unambiguous statements about the possible degree and duration of alignment:

(1) The duration of the alignment depends on the frequency differences between the involved states. A broad spectrum of state frequencies will always cause a short duration of alignment. A narrow ('cold') spectrum will cause a long alignment duration. Excitation can lead to a narrow spectrum either because the sample is cold (e.g., 0.4 K in He) and transitions selection rules limit the energy introduced by the excitation step (e.g., $\Delta J = \pm 2$ Raman selection rules), or because a narrow bandwidth excitation source is used (cf. the 1975 Heritage paper).

(2) The maximum degree of alignment depends on the number of involved states and also on their spectrum. We should think about alignment as destructive interference of transition terms from all wave packet states into the final detected state. Superposition of the states can lead to destructive interference along some spatial directions. This clarifies a theoretical limit for alignment: A higher degree of alignment requires the superposition of more quantum states. There is a direct trade-off between the number of states involved and the obtainable degree of alignment. Heritage in 1975 had lousy, but long-lasting alignment.

(3) When aiming for long-lasting and strong alignment, points (1) and (2) indicate that we must

find a compromise between (a) long lasting, or (b) strong alignment. Point (a) requires a wave packet of only the lowest frequency states for long-lasting alignment, but the alignment will be not very strong. Point (b) requires a wave packet involving many states, but the duration of the alignment is now reduced because the states span a larger frequency range.

From the theoretical viewpoint laid out above, a long-lasting and strong alignment is only possible if there are many states (-> strong alignment allowed) with very low frequencies (-> long-lasting alignment is possible). The rotational frequency reduction in helium droplets may therefore allow fundamentally better and longer-lasting alignment than possible for gas-phase species, but the improvement is directly proportional to the reduction in rotational frequencies. Note that larger molecules have inherently lower rotational frequencies and can be better aligned for longer times (at least theoretically).

To summarize, it is important to remember that the degree and duration of alignment will always directly reflect the involved quantum states (even if the experimental realization is far from the theoretical limit) and it is useful to think about the frequency spectrum and the temperature of the molecular ensemble.

A final comment on the authors modeling of I₂ alignment (supp. info.). I have no experience with modeling truncated adiabatic alignment, but if I assume that the excitation somehow resembles a Raman excitation at a 0.4 K equilibrated temperature ($\Delta J = \pm 2$ transitions from the states occupied with Boltzmann equilibrium) then I would easily expect such long-lived alignment. 0.4 K corresponds to an energy of $kT/h = 8.3$ GHz, i.e., a state with energy 8 GHz has $1/e$ population (*degeneracy) compared to the lowest rotational state. That would leave the $J=0$ and $J=2$ states as predominantly populated states for I₂, with Raman transitions up to $J=4$; difference frequencies < 20 GHz and corresponding revival periods > 50 ps. I must assume that the proper modeling of the excitation process by the authors somehow leads to population of higher rotational states (i.e., involving multiple Raman transitions), or a non-equilibrium population of rotational states? This model then clearly disagrees with the observed long rotational dispersion time and cannot teach us anything about the observed systems - but it also does not detract from our knowledge about the required superposition of rotational states that would produce alignment signals such as observed.

We agree on a number of the points brought up by the referee, in particular that with the common knowledge from gas-phase rotational wave packets, the aim of strong, field-free, long-lasting alignment may seem difficult to achieve because, at one hand, strong, field-free alignment would imply short-lasting alignment and, on the other hand, long-lasting field-free alignment would imply weak alignment. (We are, of course, talking about FIELD-FREE alignment, otherwise adiabatic alignment would trivially fulfill the criteria of very strong alignment for very long times).

We also agree that molecules in He droplets should be better suited for achieving long-lasting field-free alignment because the increased moment of inertia causes a higher density of rotational states compared to the corresponding gas-phase molecules. However, this increased moment of inertia cannot fully explain the impeding effect of the He environment on the rotational motion of the molecules. This is precisely what we quantitatively demonstrate in Sec. 2 of the Supplemental

Material. There we show that for iodine molecules in He droplets the effective moment of inertia must be increased by a factor of 10 compared to the gas-phase moment of inertia in order for a gas-phase calculation (solving the time-dependent Schroedinger equation) to match the experimental results. This is not compatible with the value of 1.7 determined by a path integral Monte Carlo calculation [see Shepperson et al., PRL 118, 203203 (2017) – including the Supplemental Material].

As such, the precise mechanism for the impeding effect is not yet fully understood. We have already addressed this in the first revision of the manuscript (page 4, line -11 – line -7) including section 2 in the Supplementary Material, where both the increased moment of inertia as well as the influence of temperature is discussed.

We do by the way, agree with the referee that angular momentum transfer from the molecules to the droplet is likely to play a role for the rotation-impeding effect and, more generally, for explaining the laser-induced rotational dynamics of the molecules inside He droplets. In collaboration with Misha Lemesko's theoretical group we are currently investigating this intensely and a picture is starting to appear. We are, however, not yet ready to make definitive statements about the fundamental mechanism for the impeding effect. Consequently, in the current manuscript we only show how the rotation-impeding effect can be exploited to create long lasting, field-free alignment, a finding that can open new opportunities in molecular science. The explanation of the impeding effect will be addressed in forthcoming manuscripts combining new theory and additional experimental results.

Comments from referee 3 in the marked up PDF:

The referee had a large number of suggestions for improvements of various sentences and formulations in our manuscript. We have implemented the majority of these suggestions and must admit that they make our text more precise and easier to understand in the relevant places. We thank the referee for her/his extraordinarily meticulous reading and constructive correction of our manuscript.

Page 4:

I am not convinced that cooling gets significantly harder with the size of the molecule. Please cite the literature if there is published work on this topic. Otherwise you might introduce factually questionable claims into the scientific literature.

We changed the sentence to:

“ . . . because the latter typically cannot be cooled to such low temperatures by supersonic jet-cooling techniques.”

and added three new references that concerns the standard way of making cold molecular beams.

Page 6:

Revivals are also observed for non-symmetric molecules. Riehn et al. show RCS spectra for p-cyclohexylaniline and cyclohexylbenzene [C. Riehn / Chemical Physics 283 (2002) 297–329], and think Felker et al. discussed similar systems. Not all those systems might show your desired degree of alignment in the revivals, but currently this sentence is hard to understand.

The sentence:

“As the technique does not depend on rotational revivals, there is no requirement for molecular symmetry.”

was included in response to a comment from referee 1. We have now changed it slightly to:

“As the technique does not depend on rotational revivals, there is no requirement for molecular symmetry other than that the polarizability must be anisotropic.”

Page 8:

It is not clear what the reader should learn from this statement. Maybe you could describe more clearly what has / has not been done to analyze the alignment dependence of SFA. Or maybe just leave it out.

We agree and decided to leave out the sentence

Page 9:

The described alignment method is still limited to molecules with a large anisotropy of polarizability. The word 'general' might be too strong here.

We changed the sentence to:

“ . . . an approach to align a broad class of large molecules inside He nanodroplets

Page 10

As explained in my statements concerning the physics of molecular alignment, there is a trade-off between the physically feasible degree of alignment and the alignment duration. Unfortunately you cannot cheat on the physics with clever experimental tricks. However, if you have multiple rotational states with very low rotational frequencies (i.e., rotation of molecule + He solvent), your situation might be better than expected.

We are not cheating on the physics. We are just suggesting a faster turn-off. We expect that the rotational dynamics will be the same following a 1 ps turn-off compared to the current aprx. 10 ps turn-off. Thus, we will gain an additional $10 - 1 \text{ ps} = 9 \text{ ps}$ field-free window.

Page 10:

Please discuss the 'interesting questions regarding strong-field physics' instead of vague referring to unspecified limits of modeling some unspecified complexity.

We have done so.

Page 10:

It is unclear to me what this sentence tries to convey.

The sentence was included in response to a comment by referee 1. We agree that the sentence was hard to follow. Consequently, it has been rewritten and moved forward in the paragraph as it fits context better there.

Page 10:

reaction dynamics (?), electronic, vibrational, rotational, ... (?)

Sorry for being unclear here. We meant "reaction dynamics". The text has been corrected.

REVIEWERS' COMMENTS:

Reviewer #3 (Remarks to the Author):

I am willing to support the publication of the manuscript in its current state. I am still not happy about the lack of a discussion or explanation of the "impeding effect" of the helium environment but the manuscript gained enough clarity that it deserves publication in its current form.

The additional self-citations ("the impeding effect of the He environment on molecular rotation observed earlier. 17,20,21") do not mention or define the word "impeding" and I believe the word serves as a pseudo-explanation for an effect the authors are unwilling to explain or discuss in a physically meaningful way. The manuscript would be significantly stronger if it would discuss the possible origin of the prolonged strong alignment. All possible explanations point to previously unexplored properties of He droplets, i.e., a high rotational state density at very low temperature / energies, or a rotational cooling and dephasing on the experimental time-scales. Even if "A theoretical description of the dynamics of laser-induced molecular alignment in helium droplets is [... yet to be] explored and developed", it might be interesting to mention the physical principles that would allow to explain this observation. Any theoretical model will show exactly (and only) those physical effects that the modeling scientist chose to include - and this will not replace the open-minded discussion of the underlying physics.

Reviewer #3 (Remarks to the Author):

I am willing to support the publication of the manuscript in its current state. I am still not happy about the lack of a discussion or explanation of the "impeding effect" of the helium environment but the manuscript gained enough clarity that it deserves publication in its current form.

The additional self-citations ("the impeding effect of the He environment on molecular rotation observed earlier. 17,20,21") do not mention or define the word "impeding" and I believe the word serves as a pseudo-explanation for an effect the authors are unwilling to explain or discuss in a physically meaningful way.

Actually, reference 20 and 21 explicitly state that the helium environment impedes the rotational motion of the molecules.

The manuscript would be significantly stronger if it would discuss the possible origin of the prolonged strong alignment. All possible explanations point to previously unexplored properties of He droplets, i.e., a high rotational state density at very low temperature / energies, or a rotational cooling and dephasing on the experimental time-scales. Even if "A theoretical description of the dynamics of laser-induced molecular alignment in helium droplets is [... yet to be] explored and developed", it might be interesting to mention the physical principles that would allow to explain this observation. Any theoretical model will show exactly (and only) those physical effects that the modeling scientist chose to include - and this will not replace the open-minded discussion of the underlying physics.

We have added a single sentence on page 4 to speculate about the physical origin of the impeding effect:

"We suspect it is related to a significant deviation of the rotational level structure from a rigid rotor structure."